# A Novel Control Method for Permanent Magnet Synchronous Linear Motor Based on Model Predictive Control and Extended State Observer

**Zhuobo Dong, Zheng Sun \***, **Hao Sun, Wenjun Wang and Xuesong Mei**

School of Mechanical Engineering, Xi'an Jiaotong University, Xi'an 710054, China;
dongzhuobo@stu.xjtu.edu.cn (Z.D.); sunhao000408@stu.xjtu.edu.cn (H.S.)
\* Correspondence: zheng.sun@xjtu.edu.cn

**Abstract:** Permanent magnet synchronous linear motor (PMSLM) is widely used to meet the requirement of high dynamic accuracy positioning, such as in machine tools and devices of semiconductor manufacturing. A new 2-DOF control structure is proposed in this paper to improve the dynamic performance of the positioning servo system with PMSLM. Aiming at the position tracking performance, a control algorithm based on the model predictive control (MPC) is developed with position and speed as the feedback state variables. In addition, an extended state observer (ESO) is designed for the rejection of various disturbances, which are not involved in the control model and are regarded as the lumped disturbance to be estimated and compensated by the ESO. The experimental results show that, compared with the commonly used PPI controller (proportional position controller and proportional–integral speed controller), the proposed method enhances the position bandwidth and servo stiffness effectively.

**Keywords:** PMSLM; MPC; ESO; dynamic performance; anti-disturbance performance

## 1. Introduction

The precision servo feed system is a relevant prerequisite for precision machining. It is necessary to improve the closed-loop bandwidth of the positioning servo control since both the tracking error and the servo stiffness are dominated by the bandwidth directly. A servo system with high bandwidth can improve the quality of machined parts and reduce the machining time. However, for the commonly used feed drives with ball screws or rack pinions, the bandwidth of the position loop is limited by the first-order resonance of these mechanical transmission systems included in the control loop [1]. Compared with the traditional feed drives with transmission systems, the "direct drive" with the permanent magnet synchronous linear motor (PMSLM) eliminates the influence of mechanical resonance on the controller fundamentally and improves the servo accuracy and dynamic response performance effectively.

The feed drives are normally cascade controlled by a proportional (P) controller in the outer position loop and a proportional–integral (PI) controller in the speed loop (PPI). The PPI controller is easy to tune and has high robustness, which meets the requirements of most industrial applications. However, its bandwidth has a lower upper limit to avoid large overshoots and oscillations. The PI-controlled speed loop can be regarded as a large inertia delay block in series. The integral term improves the performance of disturbance rejection, but the additionally introduced pole in the inner loop damages the stability reserve of the outer position control loop.

As the substitution of cascade control, the current servo control methods mainly include sliding mode control, active disturbance rejection control, adaptive control based on servo parameter estimation, robust control, etc. Sliding Mode Control (SMC) is one of the most robust algorithms with a low-accuracy system model and is insensitive to

internal and external disturbances. Extensive research has focused on the use of new sliding mode control structures [2–6] and the composite control system [7–9] combining SMC and other control methods. Most methods based on SMC have many parameters. The coupling between parameters makes SMC difficult to adjust and limits the industrial application. Active Disturbance Rejection Control (ADRC) can provide high servo stiffness for linear motors due to its property of active disturbance rejection [10]. The current research mainly focuses on improving the traditional ADRC with the compensation method [11]. Although it has a high disturbance rejection performance, parameter tuning is still a difficult problem when it is applied. Servo parameter identification and adaptive control [12], robust control [13], and other methods are also studied in positioning systems with linear motors. However, the control method combining parameter identification and adaptive control has a large amount of online calculations and requires high computational performance of the controller. Moreover, the synthesized order of $H^\infty$ controllers is normally too high to realize in industry.

At present, the research and application of Model Predictive Control (MPC) in the field of motor control have become more extensive. Kwon et al. present the principle of the MPC and analyze the stability [14]. Based on the cascade structure some researchers adopt the MPC to break through the performance limitations, such as speed/current integrated MPC controllers [15–18]. The essential difference between electromagnetic and mechanical characteristics differentiates the mechanical and electrical time constants greatly. It is not suitable for the overall control. Therefore, some researchers study to control the speed using the MPC separately. Li et al. proposed an improved predictive function for the speed regulation of the servo system with PMSM, which effectively improved the disturbance performance [19]. Wang et al. proposed an MPC controller combined with a Kalman filter, which improved the tracking performance and disturbance rejection of system speed [20]. Shao et al. presented a generalized predictive controller with a high-order sliding-mode observer [21]. Yao et al. proposed a new speed nonlinear direct predictive control method for PMSM [22]. Consequently.

The controller design should meet the requirements of the fast response to track the reference and strong robustness against the inner and outer disturbance. The traditional controlled servo systems cannot meet these two demands at the same time, so the two-degrees-of-freedom (2-DOF) control structure is applied with one controller located in the forward channel for the tracking performance and another controller or observer located in the feedback channel for the disturbance. Li et al. proposed a 2-DOF $H^\infty$ robust speed control method for the servo system [23]. The method has a good speed tracking performance and a strong robustness against load disturbance and parameter perturbation, but the design of the weight function is cumbersome. Chen et al. formed a 2-DOF controller with a fractional order Proportional-Derivative (PD) controller and an Extended State Observer (ESO) and applied it to the speed control of PMSM [24]. A 2-DOF controller proposed by Yang et al. combines a PI controller and a Kalman filter and shows a good balance between the disturbance rejection and tracking performance [25]. Until now, most research focused on the speed control of PMSM. Although the bandwidth of the position loop can also be improved with the enhancement of the speed performance, the direct controller design for the position control is seldom seen.

This paper introduces the MPC into the positioning servo system with PMSLM and proposes a high-performance positioning servo control algorithm based on the MPC and ESO. Using the MPC, a forward controller of the servo system is designed to improve the dynamic tracking performance of the system. Aiming at disturbances such as load force variation, mass perturbation, and nonlinear thrust fluctuation, an ESO is designed, which can observe and compensate in the form of lumped disturbance. The overall control structure can be depicted in Figure 1.

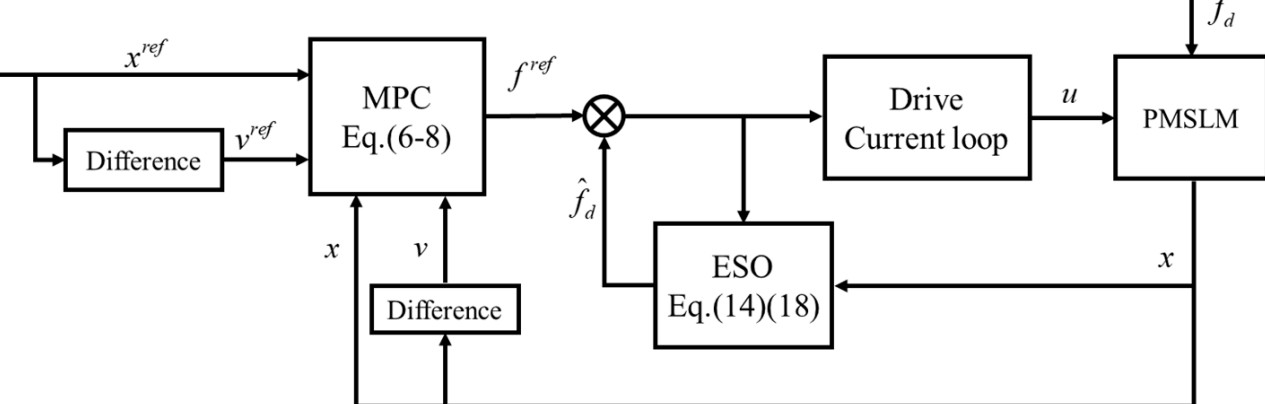

**Figure 1.** Structure of the proposed control method.

The remainder of this paper is as follows. The MPC for PMSLM is designed and analyzed in Section 2. In Section 3, the ESO for the disturbance is presented, and its pole placement and noise sensitivity are analyzed. The experimental verification is shown in Section 4. Section 5 concludes with comments.

## 2. Model Predictive Control

### 2.1. Linear Motor State Equation

Neglecting the outer disturbance force, the linear motor can be abstracted as a one-mass system. Its dynamic equilibrium equation can be formulated as

$$m \cdot \ddot{x} = f - d \cdot \dot{x} \tag{1}$$

where $x$ is the motor position, m is the total mass of the motor mover and load, $d$ is the viscose damper from the guideway, and $f$ is the electromagnetic force, which can also be regarded as the control output. It can be expressed in matrix form with the introduction of state vector $X$.

$$\dot{X} = A_c \cdot X + B_c \cdot f$$

$$\text{with } X = \begin{bmatrix} x \\ \dot{x} \end{bmatrix} \quad A_c = \begin{bmatrix} 0 & 1 \\ 0 & \frac{-d}{m} \end{bmatrix} \quad B_c = \begin{bmatrix} 0 \\ \frac{1}{m} \end{bmatrix} \tag{2}$$

Since the servo system is operated and controlled discretely, the continuous state equation should be discretized through first-order approximation

$$A = e^{A_c T_s} \approx I + A_c T_s$$

$$B = \int_0^{T_s} e^{A_c t} dt \cdot B_c \approx I T_s \cdot B_c \tag{3}$$

where $T_s$ is the servo control cycle.

According to Equation (3), the discrete state space representation can be formulated as

$$X_{k+1} = A \cdot X_k + B \cdot f_k$$

$$\text{with } X_k = \begin{bmatrix} x_k \\ v_k \end{bmatrix} \quad A = \begin{bmatrix} 1 & T_s \\ 0 & 1 - \frac{d}{m} T_s \end{bmatrix} \quad B = \begin{bmatrix} 0 \\ \frac{T_s}{m} \end{bmatrix} \tag{4}$$

The subscript $k$ means the *k-th* present moment.

### 2.2. Design of the MPC

Equation (4) transforms the state at the moment $k$ to the state of $k + 1$. Multiple using Equation (4), the state of moment $k + n$ can be predicted theoretically as follows.

$$
\begin{aligned}
X_{k+2} &= A(AX_k + Bu_k) + Bu_{k+1} \\
&= A^2 X_k + ABu_k + Bu_{k+1} \\
X_{k+3} &= A^3 X_k + A^2 Bu_k + ABu_{k+1} + Bu_{k+2} \\
X_{k+n} &= A^n X_k + \sum_{i=1}^{n} A^{i-1} Bu_{k+n-i}
\end{aligned}
\tag{5}
$$

Denoting the prediction step of the system as $n_p$, the control step as $n_c$, the state from moment $k + 1$ to the moment $k + n_p$ and $k + n_c$ can be predicted. Normally, the control step should not be beyond the prediction step $n_c \leq n_p$, and the control output is unchanged beyond the control step, $f_{k+i} = f_{k+n_c}$ with $i = n_c + 1, \cdots, n_p$. We summarize all the predicted states in a vector $Z = \begin{bmatrix} X_{k+1} & X_{k+2} & \cdots & X_{k+n_p} \end{bmatrix}^T$ and all control outputs in another vector $F = \begin{bmatrix} f_k & f_{k+1} & \cdots & f_{k+n_c-1} \end{bmatrix}^T$, the state prediction can be compactly expressed as follows.

$$
Z = M \cdot X_k + \Pi \cdot F
$$

$$
\text{with } M = \begin{bmatrix} A \\ A^2 \\ \vdots \\ A^{n_p} \end{bmatrix} \quad
\Pi = \begin{bmatrix}
B & & 0 & \cdots & 0 \\
AB & B & & & \vdots \\
\vdots & \vdots & \ddots & & 0 \\
A^{n_c-1}B & A^{n_c-2}B & & \cdots & B \\
\vdots & \vdots & \vdots & & \vdots \\
A^{n_p-1}B & A^{n_p-2}B & \cdots & & A^{n_p-n_c}B
\end{bmatrix}
\tag{6}
$$

The basic idea of the MPC of the servo system is to find a suitable $F$ to minimize the object function, which describes the difference (tracking error) between the predicted state $Z$ and the reference state $Z^{ref}$. For the servo control of PMSLM, the reference position and velocity are generated by the CNC system. Its current and future values can be easily obtained from the buffer cache. In general, to prevent the oversized control output, the electromagnetic force $F$ should also be considered in an objective function by weight. So, the object function can be constructed as follows.

$$
J = \left( Z - Z^{ref} \right)^T W_Z \left( Z - Z^{ref} \right) + F^T W_F F
$$

$$
F = \underset{F}{arg\min} J
$$

$$
\text{with } W_Z = \begin{bmatrix} w_x & & 0 \\ & w_v & \\ 0 & & \ddots \end{bmatrix}_{2n_p \times 2n_p} \quad
W_F = \begin{bmatrix} w_f \end{bmatrix}_{n_c \times n_c}
\tag{7}
$$

The weight $w_x$ and $w_v$ dominate the influence of position and speed tracking errors, respectively, and $w_f$ is weight to constrain the electromagnetic force.

Take Equation (6) into Equation (7), the minimized value of $J$ can be obtained through the following equation.

$$
\frac{\partial J}{\partial F} = 2\Pi^T W_z (MX_k + \Pi F - Z) + 2W_F F = 0
$$

$$
\Rightarrow F = \left( \Pi^T W_Z \Pi + W_F \right)^{-1} \Pi^T W_Z (Z - MX_k)
\tag{8}
$$

The obtained $F$ is a vector with the length $n_c$, but only the first value of $F$ is outputted as the current command to the motor. For the implementation of the MPC in the hardware, most coefficients can be calculated offline, such as in Equation (8), the term before $(Z - MX_k)$ is calculated offline since all the model parameters and weights are given.

### 2.3. Stability Analysis of the MPC

The control output can be reformed as the multiplication of the state error and the coefficient $K$, which is defined as follows.

$$F = K(Z - MX_k)$$
$$\text{with } K = \begin{bmatrix} 1 & 0 & \cdots & 0 \end{bmatrix}_{1 \times n_c} \left( \Pi^T W_z \Pi + W_F \right)^{-1} \Pi^T W_Z \tag{9}$$

Bringing it into Equation (4), we can obtain a close-loop recurrence relation.

$$\begin{aligned} X_{k+1} &= AX_k + BK(Z - MX_k) \\ &= (A - BKM) \cdot X_k + BK \cdot Z^{ref} \end{aligned} \tag{10}$$

According to the stability criterion, the discrete system is asymptotically stable only if all eigenvalues of $A - BKM$ are located in the unit circle.

Assuming that the weights of each step are the same, there are three weights or parameters that need to be tuned, namely, $w_x$ $w_v$, and $w_f$ correspond to the position, speed, and motor force, respectively. They describe different physical values and cannot be directly compared. To find out the suitable order of magnitude of three weights, their units should be unified first. As a reference, the weight of the electromagnetic force $w_f$ is set to be 1, the speed is the integral of the acceleration, which is the ratio of the force and the mass. Therefore, the speed weight $w_v$ should be set based on the value of $m/T_s$. Similarly, the position is the integral of velocity, so $w_x$ should be set based on the value of $m/T_s^2$.

It is difficult to solve the eigenvalue of $A - BKM$ analytically. Therefore, we calculate the maximum eigenvalue numerically by selecting the weights within a certain range. Set $n_p = 20$, $n_c = 1$, and $w_f = 1$, the norm of the maximum eigenvalue is plotted in Figure 2 by $w_x$ from 1 to $200,000 \times m/T_s^2$, and $w_v$ from 1 to $100 \times m/T_s$. The system parameters are set as the values from the test bench introduced in Section 4.

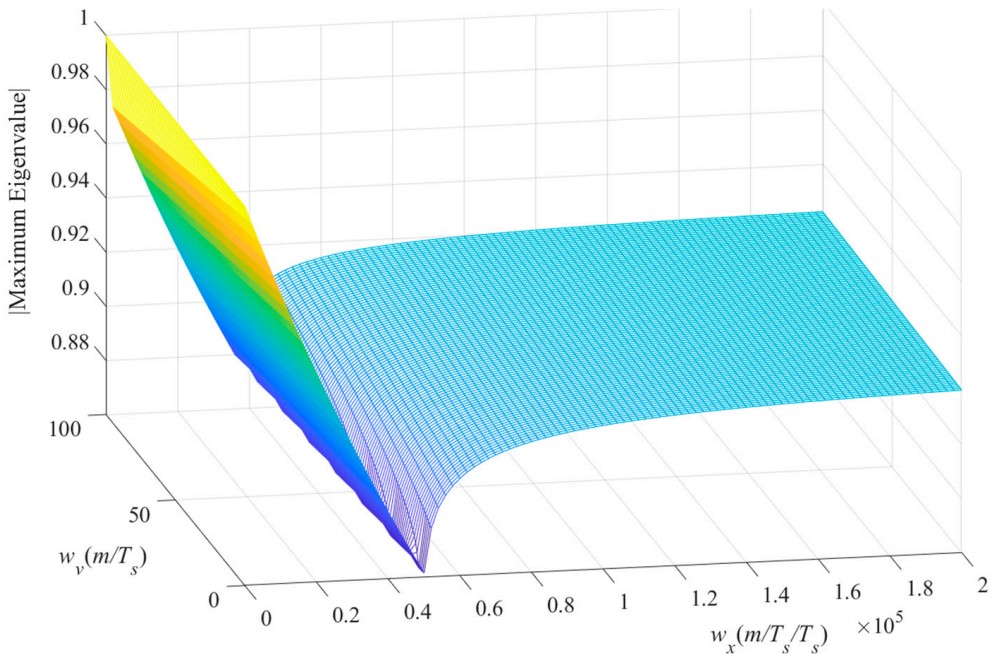

**Figure 2.** Norm distribution of maximum eigenvalue of $A - BKM$.

It can be seen from Figure 2 that all the norms of eigenvalues are smaller than one, although for some parameters they are close to one. It means that the proposed MPC has sufficient stability theoretically. However, in a real system, it is necessary to consider the bandwidth limitation of the current control loop, the current saturation, the time delay, and other factors, which damage the stability reasonably.

## 3. Extended State Observer

### 3.1. Construction of ESO

In the model of PMSLM, the disturbance forces, such as cutting force, are neglected. In addition to that, many other factors, such as the changing of the load mass, static friction, etc., also have a great impact on the control performance. The result of these factors can be equivalented in the form of a lumped disturbance. Since the uncertainty of $f_d$, it is difficult to deal with in the MPC. In this section, an extended state observer is developed to estimate the disturbance and improve the control performances.

Considering the lumped disturbance $f_d$, the dynamic equation of PMSLM can be reformulated as

$$m\ddot{x} = f_e + f_d \tag{11}$$

In Equation (11), the viscose damper can further be neglected to simplify the derivation procedure. As a new variable, the lumped disturbance force $f_d$ can be introduced into the state equation, so the extended state equation disturbing force can be constructed as follows.

$$\dot{X}_e = A_e X_e + B_e f + E \cdot \dot{f}_d$$

$$\text{with } X_e = \begin{bmatrix} x \\ \dot{x} \\ f_d \end{bmatrix} \quad A_e = \begin{bmatrix} 0 & 1 & 0 \\ 0 & 0 & \frac{1}{m} \\ 0 & 0 & 0 \end{bmatrix} \quad B_e = \begin{bmatrix} 0 \\ 1/m \\ 0 \end{bmatrix} \quad E = \begin{bmatrix} 0 \\ 0 \\ 1 \end{bmatrix} \tag{12}$$

The lumped disturbance and its derivative cannot be measured directly, it can only be estimated as a known quantity. For linear motors, the only directly measurable state is the motor position. Therefore, the lumped disturbance can be estimated by observing the actual position. The extended state observation equation is established as follows.

$$\dot{\hat{X}}_e = A_e \hat{X}_e + B_e f + GC(X_e - \hat{X}_e)$$

$$\text{with } \hat{X}_e = \begin{bmatrix} \hat{x} \\ \hat{\dot{x}} \\ \hat{f}_d \end{bmatrix} \quad G = \begin{bmatrix} g_1 \\ g_2 \\ g_3 \end{bmatrix} \quad C = \begin{bmatrix} 1 & 0 & 0 \end{bmatrix} \tag{13}$$

In this paper, the values with hut are the estimated values. The vector G is the gain vector including three gains for the observer. The vector C is the output vector denoting that only the position can be directly measured and compared with the estimated value.

Similar to Equation (3), the disturbance observation is applied discretely. So, Equation (13) is discretized through the Tyler expansion.

$$\hat{X}_{e,k+1} = A_{ed}\hat{X}_{e,k} + B_{ed}f_k + G_d(CX_{e,k} - C\hat{X}_{e,k})$$

$$\text{with } \hat{X}_{e,k} = \begin{bmatrix} \hat{x}_k \\ \hat{\dot{x}}_k \\ \hat{f}_{d,k} \end{bmatrix} \quad A_{ed} = \begin{bmatrix} 1 & T_s & \frac{T_s^2}{2m} \\ 0 & 1 & \frac{T_s}{m} \\ 0 & 0 & 1 \end{bmatrix} \quad B_{ed} = \begin{bmatrix} \frac{T_s^2}{2m} \\ \frac{T_s}{m} \\ 0 \end{bmatrix} \quad G_d = \begin{bmatrix} g_1 T_s + g_2 \frac{T_s^2}{2} \\ g_2 T_s + g_3 \frac{T_s^2}{2m} \\ g_3 T_s \end{bmatrix} \tag{14}$$

### 3.2. Gains Tuning and Stability Analysis

The tuning and analysis in this section are still based on the continuous observation process. Subtracting Equation (12) into Equation (13), the estimation error can be obtained.

$$\dot{\zeta} = (A_e - GC)\zeta - E\dot{f}_d$$

$$\Rightarrow \begin{bmatrix} \dot{\varepsilon}_1 \\ \dot{\varepsilon}_2 \\ \dot{\varepsilon}_3 \end{bmatrix} = \left( \begin{bmatrix} 0 & 1 & 0 \\ 0 & 0 & \frac{1}{m} \\ 0 & 0 & 0 \end{bmatrix} - \begin{bmatrix} g_1 \\ g_2 \\ g_3 \end{bmatrix} \begin{bmatrix} 1 & 0 & 0 \end{bmatrix} \right) \begin{bmatrix} \varepsilon_1 \\ \varepsilon_2 \\ \varepsilon_3 \end{bmatrix} - \begin{bmatrix} 0 \\ 0 \\ 1 \end{bmatrix} \dot{f}_d \tag{15}$$

where $\varepsilon_1$, $\varepsilon_2$ and $\varepsilon_3$ denote the estimation error of position, speed, and disturbance, respectively.

The observer gains were designed through the pole placement method. If the observation process is stable, the eigenvalues of $A_e - GC$ must be located in the left half plane of the s-plane. We can set that when this third-order matrix has a triple negative real pole $-\omega_0$, the characteristic polynomial of $A_e - GC$ should on one hand satisfy the following equation with the cut-off frequency $\omega_0$.

$$(s + \omega_0)^3 = s^3 + 3\omega_0 s^2 + 3\omega_0^2 s + \omega_0^3 \tag{16}$$

On the other hand, the eigenvalue is calculated through the determinant.

$$det(sI - A_e + GC) = \begin{vmatrix} s + g_1 & -1 & 0 \\ g_2 & s & -1/m \\ g_3 & 0 & s \end{vmatrix} = s^3 + g_1 s^2 + g_2 s + \frac{g_3}{m} \tag{17}$$

Comparing the coefficients of Equation (16) and Equation (17), the gain vector can be valued as follows.

$$g_1 = 3\omega_0$$

$$g_2 = 3\omega_0^2 \tag{18}$$

$$g_3 = m\omega_0^3$$

### 3.3. Noise Sensitivity Analysis of ESO

The noise in the real system cannot be avoided. The observer with unsuitable parameters may enlarge the noise and bring the whole system into instable. Therefore, noise sensitivity will be analyzed in this section. Introducing the position measuring noise $\eta$ into Equation (13), the observation equation can be extended as

$$\begin{bmatrix} \dot{\hat{x}} \\ \dot{\hat{x}} \\ \dot{\hat{f}}_d \end{bmatrix} = \begin{bmatrix} 0 & 1 & 0 \\ 0 & 0 & \frac{1}{m} \\ 0 & 0 & 0 \end{bmatrix} \begin{bmatrix} \hat{x} \\ \hat{x} \\ \hat{f}_d \end{bmatrix} + \begin{bmatrix} 0 \\ \frac{1}{m} \\ 0 \end{bmatrix} f_e + \begin{bmatrix} g_1 \\ g_2 \\ g_3 \end{bmatrix} \begin{bmatrix} 1 & 0 & 0 \end{bmatrix} \left( \begin{bmatrix} x + \eta \\ \dot{x} \\ f_d \end{bmatrix} - \begin{bmatrix} \hat{x} \\ \hat{x} \\ \hat{f}_d \end{bmatrix} \right) \tag{19}$$

Subtracting Equation (12) from Equation (19), the estimation error can be obtained as follows

$$\begin{bmatrix} \dot{\varepsilon}_1 \\ \dot{\varepsilon}_2 \\ \dot{\varepsilon}_3 \end{bmatrix} = \left( \begin{bmatrix} 0 & 1 & 0 \\ 0 & 0 & \frac{1}{m} \\ 0 & 0 & 0 \end{bmatrix} - \begin{bmatrix} g_1 \\ g_2 \\ g_3 \end{bmatrix} \begin{bmatrix} 1 & 0 & 0 \end{bmatrix} \right) \begin{bmatrix} \varepsilon_1 \\ \varepsilon_2 \\ \varepsilon_3 \end{bmatrix} - \begin{bmatrix} 0 \\ 0 \\ 1 \end{bmatrix} \dot{f}_d + \begin{bmatrix} g_1 \\ g_2 \\ g_3 \end{bmatrix} \eta \tag{20}$$

With the error Equation (20) and the value of G determined by Equation (18), we can obtain the transfer function from $\eta$ to $\varepsilon_3$

$$\frac{\varepsilon_3(s)}{\eta(s)} = \frac{m\omega_0^3 s^2}{(s + \omega_0)^3} \tag{21}$$

The amplitude of the transfer function can be calculated as

$$\left| \frac{\varepsilon_3(j\omega)}{\eta(j\omega)} \right| = \frac{m\omega^2\omega_0^3}{(\omega^2 + \omega_0^2)^{3/2}} \tag{22}$$

It is a monotonic increasing function about the cut-off frequency $\omega_0$. Increasing $\omega_0$, the amplitude from $\eta$ to $\varepsilon_3$ is enlarged [26], which means that the jitter equivalent to the estimated disturbance is also increased.

## 4. Experimental Verification

### 4.1. Setup of Test Bench

The test bench for the verification was built with a direct drive platform from CNBHC, as shown in Figure 3. The applied linear motor type was TMLA0070-095-000, and its specific parameters are listed in Table 1.

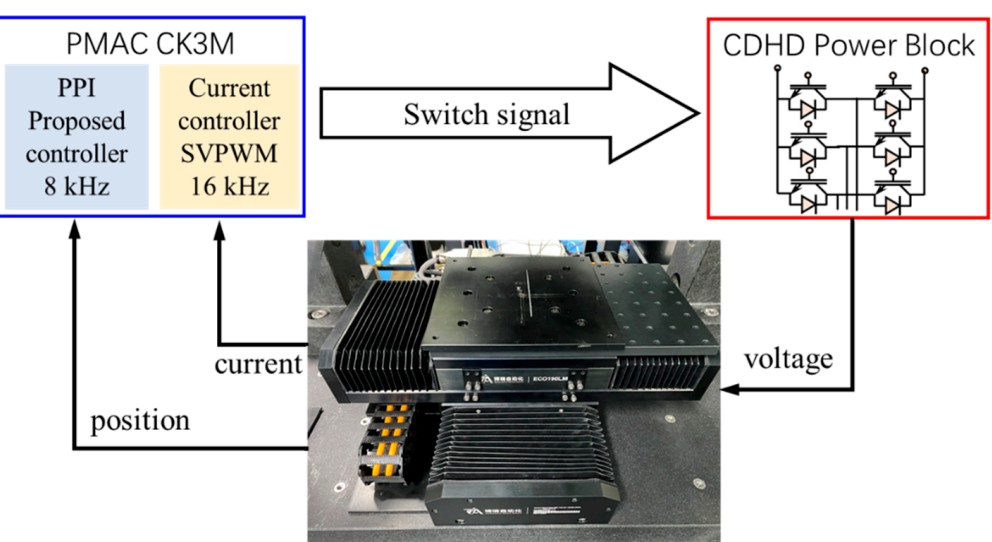

**Figure 3.** Test bench applied for the experiments.

**Table 1.** TMLA0070-095-000 linear motor parameters.

| Parameters | Parameter Value | Parameters | Parameter Value |
|---|---|---|---|
| $R/\Omega$ | 2.8 | $L_d$, $L_q$/mH | 6.8 |
| $I_{\max}/\mathrm{A}$ | 9.5 | $I_N/\mathrm{A}$ | 2.5 |
| $U_{dc}/\mathrm{V}$ | 300 | $K_m/\mathrm{V\cdot m^{-1}\cdot s^{-1}}$ | 21.4 |
| $K_e/\mathrm{N\cdot A^{-1}}$ | 32 | m/kg | 6 |

In the test bench, the linear motor was controlled by a PMAC controller with the type of CK3M from OMRON. The motor was driven by a Power Block from CDHD through the Pulse Width Modulation (PWM) signals. The current controller and PWM algorithm were implemented in CK3M with a control frequency of 16 kHz. The power block received only the switch signals for the inverter, whose frequency was the same as the current control frequency 16 kHz. The current controller applied in CK3M was a PI controller ($k_{i,p}(1 + k_{i,i}/s)$). After the auto-tuning, the control gains were parameterized as $k_{i,p} = 35\ V/A\ k_{i,i} = 411\ \mathrm{s}^{-1}$. The bandwidth of the closed current loop was over 1 kHz.

The presented servo control algorithm was implemented in CK3M through the Power PMAC IDE with a control frequency of 8 kHz. Following the trial and tuning, the weight coefficients for the proposed MPC are listed in Table 2. The prediction step $n_p$ is 20 and the control step $n_c$ is 1. With these parameters, the linear motor has a good dynamic

performance, rare current saturation, and is robust against the parameter perturbation. For comparison, a conventional PPI controller was also implemented in the CK3M. The control gain can also be seen in Table 2.

**Table 2.** Control parameters of the MPC and PPI.

| MPC | | PPI | | |
|---|---|---|---|---|
| Weight Coefficient | Value | Control Gain | Value | Unit |
| $w_x$ | $35,000 \times m/T_s^2$ | $k_{x,p}$ | 300 | rad/s |
| $w_v$ | $10 \times m/T_s$ | $k_{v,p}$ | 240 | As/m |
| $w_f$ | 1 | $k_{v,i}$ | 200 | rad/s |

*4.2. Experimental Verification of Tracking Performance*

The setup response is applied to test the tracking performance when the position commends stepwise changes from the initial position of 0 mm to 0.1 mm. For P-PI control, the MPC and MPC + ESO with different poles, the experimental results are shown and compared in Figure 4.

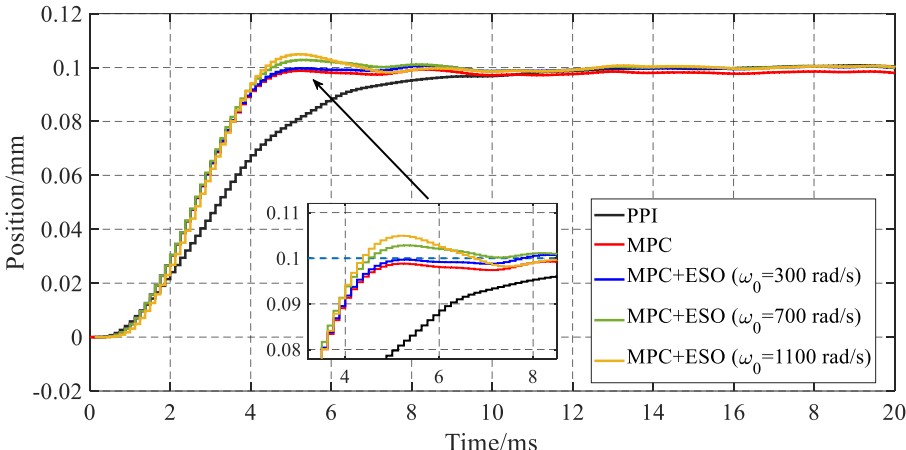

**Figure 4.** Step response of PPI, MPC, MPC + ESO.

It is obviurs that the tracking performance of the MPC is better than that of PPI control. The settling time, in which the feedback position drives in the 97% of step reference, reduced from 10.3 ms of PPI over 56% to 4.5 ms of the MPC. Due to the existence of static friction in the guide, the MPC without ESO has a certain steady-state error, so an additional anti-disturbance method is necessary. The introduced ESO eliminates the steady-state error in step response. But the overshoot can also be seen in Figure 4 when the pole value of the ESO increases. The main reason is that various disturbances and uncertainties in the real system make the ESO estimate all these factors in the form of lumped disturbance. With the increase in the cut-off frequency (the placed pole of ESO), the high-frequency components of the disturbance are involved in the estimated results, which enlarge the instantaneous change in compensation output, consequently. Although a slight overshoot can be seen here, the motor position converges to the steady-state value in a short time, so it can still be considered that the tracking performance is basically dominated by the MPC.

*4.3. Experimental Verification of Anti-Disturbance Performance*

The anti-disturbance performance is tested by applying a stepwise disturbance current of 2.5 A in front of the current loop when the motor is in a steady state with the command position of 0 mm. The experimental results of PPI control and the MPC with different cut-off frequencies of ESO are shown in Figure 5a. The estimated disturbances as the output of the ESO are shown in Figure 5b.

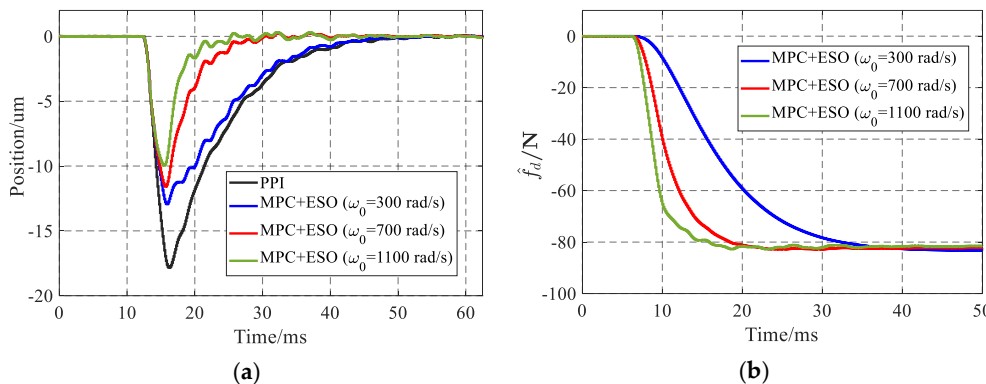

**Figure 5.** (**a**): disturbance rejection of MPC + ESO and P-PI; (**b**): estimated disturbance force from ESO.

Generally, the proposed control method shows a better disturbance rejection than the PPI control. The ESOs with different poles estimate disturbance current as the lumped disturbance and converge value of 83 N. The higher the cutoff frequency is set, the quicker the ESO output converges. Consequently, both the maximal position error and settling time decrease with the increase in the cut-off frequency. The results of different control methods are listed in Table 3 for a better comparison. Compared with the ESO with the cutoff frequency of 300 rad/s, the ESO with the cut-off frequency of 700 rad/s shows a significant enhancement of the disturbance performance with the improvement of the maximal error and the settling time of 17.8 μm and 35.7 ms from PPI to 11.6 μm and 18.1 ms, respectively. For the further increase in the cut-off frequency, the anti-disturbance performance is still improved but with a limited decline from 11.6 μm and 18.1 ms to 10.0 μm and 12.8 ms, respectively.

**Table 3.** Position error and settling time of the MPC + ESO and PPI.

| Controller | Position Error/μm | Settling Time/ms |
|:---:|:---:|:---:|
| PPI | 17.8 | 35.7 |
| MPC + ESO ($\omega_0 = 300$ rad/s) | 12.9 | 35.8 |
| MPC + ESO ($\omega_0 = 700$ rad/s) | 11.6 | 18.1 |
| MPC + ESO ($\omega_0 = 1100$ rad/s) | 10.0 | 12.8 |

Considering the noise in the feedback signal, the cutoff frequency of the ESO cannot be set with an excessively high value. Figure 6 shows the effect of the noise with different pole places on the estimation output when the motor is in the steady state of 0 mm command position. When the pole of the ESO is located at $-700$ rad/s, the jitter of the estimated disturbance is about 0.05 N, but when it is increased to $-3000$ rad/s, the jitter increases significantly to about 1 N. The trend matches the theoretical analysis of Equation (22).

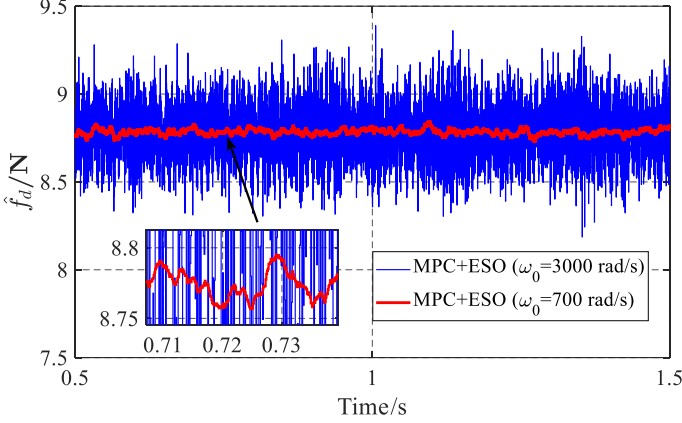

**Figure 6.** Jitter of the estimated disturbance.

### 4.4. Experimental Verification in the Frequency Domain

The effectiveness of the presented control method can be seen not only in the time domain but also in the frequency domain, as shown in Figure 7. The logarithmic frequency sweep signal is generated as the position reference with an amplitude of 0.03 mm and a frequency range from 1 Hz to 300 Hz. The close-loop bandwidth is normally defined as the frequency when the magnitude once attenuated to −3 dB. The bandwidth of the positioning system controlled by the PPI method is about 72 Hz. Using the presented control method, the bandwidth can be increased by 94.4% to over 140 Hz.

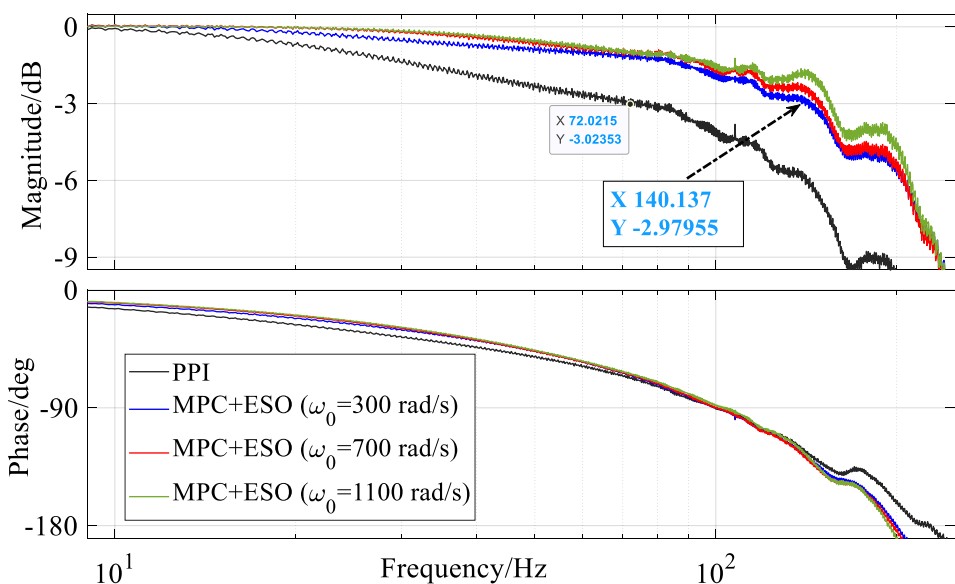

**Figure 7.** Close-loop bode plot of the MPC + ESOF and PPI.

As mentioned above, the frequency response for the position tracking is mainly dominated by the MPC, but the cut-off frequency of the observer has a certain impact on it. Increasing the cut-off frequency from 300 to 700, the property of magnitude in the middle-frequency range of about 40 Hz will be better with less attenuation, which means less transfer distortion in this frequency range. But when further increasing the cut-off frequency, a peak at about 135 Hz will be obvious. This is mainly caused by the amplify effect of the ESO on the measurement noise, as shown in Figure 7 green, and will damage the stable reverse. Therefore, choosing a suitable cut-off frequency of the ESO is relevant for industrial applications.

## 5. Conclusions

In this paper, a control method for the positioning servo system with PMSLM is proposed. It belongs to the 2-DOF control structure and consists of an extended state observer, which estimates and compensates for the lumped disturbance, and a model predictive controller dealing with the tracking performance through the feedback position and speed. Compared with the existing servo controllers, which mainly use the current state and reference value, the MPC predicts the future state, compares with the reference sequence, and solves the optimized control output. The enhanced tracking performance is expected since more state information at different times is involved. The two modules combine the control ideas of prediction, estimation, compensation, and improve the overall performance of the positioning servo system effectively.

The tuning of the MPC and ESO can be carried out separately. Although the derivation of the MPC and ESO is complex, only three parameters need to be tuned (the weight of position error $w_x$, speed error $w_v$, and the cut-off frequency of ESO $\omega_0$). The experimental results show that, compared with the mainstream PPI cascade control structure, the

designed control method has higher servo bandwidth and servo stiffness. The implementation of the proposed control method has a relatively low computation requirement for the hardware. It is suitable for application as a substitution for the standard PPI controller.

**Author Contributions:** Conceptualization, Z.S.; methodology, Z.D.; software, Z.D.; validation, Z.D. and H.S.; formal analysis, Z.S.; investigation, Z.D.; writing—original draft preparation, Z.D.; writing—review and editing, Z.S. and H.S.; visualization, Z.D.; supervision, X.M.; project administration, W.W.; funding acquisition, W.W. All authors have read and agreed to the published version of the manuscript.

**Funding:** This research was funded by the National Key Research and Development Program of China under grant number 2021YFF0500203 and funded by the National Natural Science Foundation of China under grant number 51975461.

**Data Availability Statement:** The topic of this paper is not data research of data analyse. The CNBHC dose not want to publish the experiment raw data.

**Acknowledgments:** The authors would like to thank CNBHC Co., Ltd. Chengdu, China for providing the high-precision linear motor and Mo Sang from OMRON Industrial Automation (China) Co., Ltd. Beijing branch for the technical support.

**Conflicts of Interest:** The authors declare no conflicts of interest.

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
