# Peer review of "A Novel Control Method for Permanent Magnet Synchronous Linear Motor Based on Model Predictive Control and Extended State Observer"

_actuators, doi:10.3390/act13010034_

Round 1

Reviewer 1 Report

Comments and Suggestions for Authors

This paper proposed a control method for the positioning servo system with PMSLM.

The reviewer would like to leave some comments for improving the paper. 

1. It will be good to show the effect of model uncertainties. The following reference would be helpful for model parameter estimations :

"Adaptive Torque Estimation for an IPMSM with Cross-Coupling and Parameter Variations" Energies

2. What is the sampling time used, T_s ?  Some computation burden is required for computing MPC.  How can the authors compute MPC on the hardware system used?

3. The stability of MPC should be more elabrated. The following reference would be helpful for proving the stability:

Receding Horizon Control: Model Predictive Control for State Models

 written by Wook Hyun Kwon, Soo Hee Han

It is recommended to revise and resubmit the paper if the issues above are resolved. 

Author Response

This paper proposed a control method for the positioning servo system with PMSLM. The reviewer would like to leave some comments for improving the paper.

  1. It will be good to show the effect of model uncertainties. The following reference would be helpful for model parameter estimations: "Adaptive Torque Estimation for an IPMSM with Cross-Coupling and Parameter Variations" Energies

Thank you for the suggestion. This paper focuses on servo control of PMSLM, which mainly includes position and speed control as seen in Eq. (2). The force to be estimated by the ESO is the outer disturbance force such as the cutting force of machine tool, nonlinear friction, etc. The electromagnetic model of the linear motor as well as its uncertainty is not the topic of this paper.

  1. What is the sampling time used, T_s ? Some computation burden is required for computing MPC. How can the authors compute MPC on the hardware system used?

The sampling time or the control period for the MPC and ESO is 125 us, sampling time for the current control and the SVPWM is 62.5 us. In the PMAC-CK3M, a commonly used PI controller is applied for the current control, its structure cannot be changed. But for the servo control, the user has the opportunity to implement its own controller in CK3M. Although the derivation of MPC and ESO is complex, most coefficients can be calculated offline, such as in the control law of Eq. (8), the term before (Z-MX_k) is offline calculated. The implementation in the hardware involves only multiplication and addition.

  1. The stability of MPC should be more elaborated. The following reference would be helpful for proving the stability: Receding Horizon Control: Model Predictive Control for State Models written by Wook Hyun Kwon, Soo Hee Han

Thank you very much, the suggested book is very helpful. In this paper, the stability of the MPC is analyzed through the general criterion for the discrete system, all the poles (eigenvalues) of the close loop discrete system should be in the unit circle. Since the system has only two state values (position and speed) has no integral part, it is easy to keep the system into stabile.

Reviewer 2 Report

Comments and Suggestions for Authors

In this manuscript, the authors have introduced a composite controller integrating a model predictive controller with a linear extended state observer (ESO) aimed at enhancing the position control of the permanent magnet synchronous linear motor (PMSLM). To further refine the manuscript, the following issues should be addressed:

1)      Literature Review and Motivation: It has been extensively established that integrating a feedback controller with a disturbance observer to form a composite controller can enhance several vital control performance indices. To date, a variety of composite controllers have been developed for robust position control of the PMSLM, including those based on the predictive controller. The authors should thoroughly review the evolution of composite controllers using predictive controllers for the PMSLM. Moreover, the authors should also provide a detailed discussion on the motivation for developing the presented composite controller, considering the existing landscape of composite controllers that employ predictive controllers and disturbance observers for PMSLM. This should include a critical analysis of how the proposed approach advances or differs from existing methodologies.

2)      The authors should add a block diagram to describe the principle of the used model predictive controller in detail.

3)      It is imperative for the authors to provide the full names of abbreviations at their first occurrence to aid reader comprehension. This includes ‘PPI’ in both the abstract and main body, as well as ‘PD’, ‘CNBHC’, ‘PWAC’, and ‘PWM’ in the main body of the manuscript.

4)      The switching frequency of the inverter used in experiments should be given.

5)      The expressions of the PPI controller should be given, and the parameters of the current control loop should be provided.

6)      The parameters of the PPI controller and the proposed composite controller should be presented in table.

7)      The authors should detail the methodology for selecting the parameters of the PPI controller used in the experiments. The authors must ensure and provide justification that the parameters selected for the PPI controller are optimal.

Comments on the Quality of English Language

Moderate editing of English language is required.

Author Response

In this manuscript, the authors have introduced a composite controller integrating a model predictive controller with a linear extended state observer (ESO) aimed at enhancing the position control of the permanent magnet synchronous linear motor (PMSLM). To further refine the manuscript, the following issues should be addressed:

1) Literature Review and Motivation: It has been extensively established that integrating a feedback controller with a disturbance observer to form a composite controller can enhance several vital control performance indices. To date, a variety of composite controllers have been developed for robust position control of the PMSLM, including those based on the predictive controller. The authors should thoroughly review the evolution of composite controllers using predictive controllers for the PMSLM. Moreover, the authors should also provide a detailed discussion on the motivation for developing the presented composite controller, considering the existing landscape of composite controllers that employ predictive controllers and disturbance observers for PMSLM. This should include a critical analysis of how the proposed approach advances or differs from existing methodologies.

As you mentioned there are many research on the 2-DOF controller, but all of them focuse on the current or speed tracking performance. The 2-DOF controller design for positioning systems with linear motor has seldom seen until now. We believe that the introducing of such a controller direct for the postion control can enhance the accuracy of dynamic performance of a servo system. The experimental results show the effectiveness and the advantages of the proposed method comparing with PPI controller, which is commonly used in the industry.

Also we improve the first and the last section to present the motivation and the prospect.

2) The authors should add a block diagram to describe the principle of the used model predictive controller in detail.

The control structure is shown in Fig. 1, which presents a general structure and the equations used in the block. Unlike the conventional cascade PID controller, whose detail can be easily depicted by the block diagram, the MPC is presented more suitable by the formulas, since it does not output feedback but state feedback with complex calculations for the multi-step prediction. Therefore, we think the scale of Fig. 1 is suitable.

3) It is imperative for the authors to provide the full names of abbreviations at their first occurrence to aid reader comprehension. This includes ‘PPI’ in both the abstract and main body, as well as ‘PD’, ‘CNBHC’, ‘PWAC’, and ‘PWM’ in the main body of the manuscript.

We added the full names of abbreviations at their first occurrence in the main body of the manuscript.

PMAC is a kind of controller developed by the company Delta Tau and now is sold by Omron. CNBHC is a company providing the liner motor for research. It is just called CNBHC, and has no full name.

4) The switching frequency of the inverter used in experiments should be given.

The switching frequency is the same as the control frequency of the current loop 16 kHz, and has been added in the manuscript.

5) The expressions of the PPI controller should be given, and the parameters of the current control loop should be provided.

The value of the PPI controller is added in the manuscript. The current controller is auto-tuned by the PMAC with the value of control gain Kip=35 V/A Kii=400 1/s . The bandwidth of the closed current loop is over 1000 Hz.

6) The parameters of the PPI controller and the proposed composite controller should be presented in the table.

Corrected

7) The authors should detail the methodology for selecting the parameters of the PPI controller used in the experiments. The authors must ensure and provide justification that the parameters selected for the PPI controller are optimal.

The cascade PPI controller means a proportional-integral (PI) controller in the speed loop and a proportional (P) controller in the outer position loop. The PI speed controller is tuned based on the Ziegler–Nichols method. The step-response of speed has an overshoot of less than 20% but no oscillation and static tracking error. The proportional position controller is tuned based on the industrial convention with an amplitude reserve of 10 dB.

Round 2

Reviewer 1 Report

Comments and Suggestions for Authors

All reviewer's comments are well reflected on the revised paper. 

This paper can be accepted. 

Reviewer 2 Report

Comments and Suggestions for Authors

I have no further comments.

Comments on the Quality of English Language

Minor editing of English language required.